# Choosing between AR(1) and VAR(1) models in typical psychological applications

**Fabian Dablander[1]◉\*, Oisín Ryan[2]◉, Jonas M. B. Haslbeck[1]◉**

**1** Department of Psychological Methods, University of Amsterdam, Amsterdam, Netherlands, **2** Department of Methodology and Statistics, Utrecht University, Utrecht, Netherlands

◉ These authors contributed equally to this work.
\* dablander.fabian@gmail.com

## Abstract

Time series of individual subjects have become a common data type in psychological research. The Vector Autoregressive (VAR) model, which predicts each variable by all variables including itself at previous time points, has become a popular modeling choice for these data. However, the number of observations in typical psychological applications is often small, which puts the reliability of VAR coefficients into question. In such situations it is possible that the simpler AR model, which only predicts each variable by itself at previous time points, is more appropriate. Bulteel et al. (2018) used empirical data to investigate in which situations the AR or VAR models are more appropriate and suggest a rule to choose between the two models in practice. We provide an extended analysis of these issues using a simulation study. This allows us to (1) directly investigate the relative performance of AR and VAR models in typical psychological applications, (2) show how the relative performance depends both on *n* and characteristics of the true model, (3) quantify the uncertainty in selecting between the two models, and (4) assess the relative performance of different model selection strategies. We thereby provide a more complete picture for applied researchers about when the VAR model is appropriate in typical psychological applications, and how to select between AR and VAR models in practice.

**Data Availability Statement:** All data are available from the Github archive at https://github.com/jmbh/ARVAR.

**Funding:** Fabian Dablander was supported by the NWO Vici grant 016.Vici.170.083. Oisín Ryan was

## Introduction

Time series of individual subjects have become a common data type in psychological research since collecting them has become feasible due to the ubiquity of mobile devices. First-order Vector Autoregressive (VAR) models, which predict each variable by all variables including itself at the previous time point, are a natural starting point for the analysis of dependencies across time in such data and are already used extensively in applied research [1–5].

A key question that arises when using these models is: how reliable are the estimates of the single-subject VAR model, given the typically short time series in psychological research (i.e., $n \in [30, 200]$)? To be more precise, we would like to know how large the *estimation error* is in this setting. Estimation error is defined as the distance between the estimated parameters and the parameters in the true model, assuming that the true model has the same parametric form

supported by a grant from the Netherlands Organisation for Scientific Research (NWO; Onderzoekstalent Grant 406-15-128). Jonas Haslbeck was supported by the Eropean Research Council Consolidator Grant no. 647209.

**Competing interests:** The authors have declared that no competing interests exist.

as the estimated model. If estimation error is large, it might be possible to obtain a smaller estimation error by choosing a simpler model, even though it is less plausible than the more complex model [6]. A possible simpler model in this setting is the first-order Autoregressive (AR) model, in which each variable is predicted only by itself at the previous time point. While the AR model introduces a strong bias by setting all interactions *between* variables to zero, it can have a lower estimation error when the number of available observations is small. When analyzing time series in psychological research it is therefore important to know (a) in which settings the AR or the VAR model has a lower estimation error, and (b) how to choose between the two models in practice.

Bulteel et al. [7] identified these important and timely questions, and offered answers to both. They investigated question (a) regarding the relative performance of AR and VAR models by selecting three empirical time series data sets, each consisting of a number of individual time series with the same data structure. For each of these data sets, they approximate the out-of-sample prediction error with out-of-bag cross-validation error for both the AR and the VAR model and their mixed model versions. The authors make a valuable contribution by assessing which of the many cross-validation schemes available for time series approximates prediction error best in this context. Using the approximated prediction error obtained via cross-validation, they find that the prediction error for AR is smaller than for VAR, and that the prediction error of mixed AR and mixed VAR is similar. In a last step, they link prediction and estimation error by stating that "[. . .] the number of observations $T$ [here $n$] that is needed for the VAR to become better than the AR is the same for the prediction MSE [mean squared error] as well as for the parameter accuracy [estimation error]" [7, p. 10]. Although the latter statement implies that the estimation error of mixed AR and mixed VAR models are similar, Bulteel et al. [7] conclude that "[. . .] it is not meaningful to analyze the presented typical applications with a VAR model" (p. 14) when discussing both mixed effects (i.e., multilevel models with random effects) and single-subject models.

Using their statement about the link between prediction error and estimation error together with a preference towards parsimony, Bulteel et al. [7] also offer an answer to question (b) on how to choose between the AR and VAR models in practice: they suggest using the "1 Standard Error Rule", according to which one should select the AR model if its prediction error is not more than one standard error above the prediction error of the VAR model, and select the model with lowest prediction error otherwise [8, p. 244].

In this paper, we provide an extended analysis of the problems studied by Bulteel et al. [7]. First, regarding question (a) on the relative performance of the AR and VAR models: when the goal is to determine the estimation error in a given setting, one can obtain it directly with a simulation study. A simulation study allows for a more extensive analysis of this problem for three reasons. First, we do not need to make any claim about the relation between prediction error and estimation error, which—as we will show—turns out to be non-trivial. Second, in a simulation study we can average over sampling variance which allows us to compute the expected value of estimation (and prediction) error. While the approach of Bulteel et al. [7] in using three empirical datasets has the benefit of ensuring the models considered mirror data from psychological applications, these empirical datasets are naturally subject to sampling variation. And third, a simulation study allows us to map out the space of plausible VAR models and base our conclusions on this large set of VAR models instead of the VAR models estimated from the three data sets used by Bulteel et al. [7]. We perform such a simulation study, which allows us to give a direct answer to the question of how large the estimation errors of AR and VAR models are in typical psychological applications.

Regarding question (b) on choosing between AR and VAR models in practice, Bulteel et al. [7] base their "1 Standard Error Rule" (1SER) on the idea that the $n$ at which the estimation

errors of the AR and VAR models are equal is (approximately) the same $n$ at which the prediction errors of those models are equal, combined with a preference towards the more parsimonious model. While the 1SER is used as a heuristic in the statistical learning literature [8], it is not clear whether this heuristic would perform better in the present problem than simply selecting the model with the lowest prediction error. We show that when choosing between AR and VAR models, the $n$ at which the prediction errors become equal is not necessarily the same as the $n$ at which estimation errors become equal: in fact, there is a substantial degree of variation in how the prediction and estimation errors of both models cross. Using the relationship between estimation and prediction error we are able to show via simulation when the 1SER is expected to perform better than selecting the model with lowest prediction error. This extended analysis of the problem studied by Bulteel et al. [7] provides a more complete picture for applied researchers about when the VAR model is appropriate in typical psychological applications, and how to select between AR and VAR models in practice.

## When does VAR outperform AR?

In this section we report a simulation study which directly answers the question of how large the estimation errors of AR and VAR models are in typical psychological applications. This allows the reader to get an idea of how many observations $n_e$ one needs, on average, for the VAR model to outperform the AR model. In addition, we will decompose the variance around those averages in sampling variation and variation due to differences in the VAR parameter matrix $\Phi$. Finally, explaining the latter type of variation allows us to obtain $n_e$ conditioned on characteristics of $\Phi$. The analysis code for the simulation study is available from https://github.com/jmbh/ARVAR.

### Simulation setup

Since the AR model is nested under the more complex VAR model, we focus solely on the VAR as the true data-generating model. To obtain realistic VAR models, we use the following approach: first, we estimate a mixed VAR model to the "MindMaastricht" data [9], which consists of 52 individual time series with on average $n = 41$ measurements on $p = 6$ variables, and is the only publicly available data set used by Bulteel et al. [7]. In a second step, we sample stationary VAR models with a diagonal error covariance matrix from this mixed model.

We expect that the estimation (and prediction) errors of the AR and VAR model depend not only on the number of observations $n$, but also on the characteristics of the underlying $p \times p$ VAR model matrix $\Phi$. We therefore stratify the sampling process from the mixed model by two characteristics of $\Phi$. This procedure allows us to obtain a better picture of how the performance of AR and VAR may differ depending on the characteristics of the data generating model.

The first characteristic is based on the size of the auto-regressive effects, that is, the absolute values of the diagonal elements of the lagged parameter matrix ($\Phi_{ii}$) which encode the relationship between a variable and itself at the next time point. We summarize the information contained in these diagonal elements by taking the mean of their absolute values $D$, given as

$$D = \frac{1}{p}\sum_{i=1}^{p}|\Phi_{ii}| \ .$$

Note here that taking the sum of auto-regressive parameters is equivalent to taking the sum of the *eigenvalues* of $\Phi$, denoted $\lambda$. To ensure stationarity, only $\Phi$ matrices with $|\lambda| < 1$ are included in our analysis [10]. The second characteristic is based on the size of the cross-lagged

parameters ($\Phi_{ij}$, $i \neq j$), encoding the relationships between different processes. We again summarize this information by taking the mean absolute of these parameters, denoted $O$ and given as

$$O = \frac{1}{p(p-1)} \sum_{i=1}^{p} \sum_{j \neq i}^{p} |\mathbf{\Phi}_{ij}| \quad .$$

We expect that true VAR models with a high $D$ value and small $O$ value (i.e., large auto-regressive effects and small cross-lagged effects) result in a low estimation error for AR models, since these VAR models are very similar to an AR model. In contrast, if $O$ is high, we expect that the estimation error of the AR model is large, because it sets the large cross-lagged effects in the true VAR model to zero.

Ideally, we would stratify by sampling a fully crossed grid of $D$ and $O$ values. However, this is not possible since some combinations have an extremely small probability: For example, if a matrix has auto-regressive parameters close to one, it is unlikely to describe a stationary process if it also contains high positively-valued cross-lagged parameters. We therefore adopt the following approach: we divided the $D$-$O$-space in a grid by dividing each dimension into 15 equally spaced intervals (see S1 Fig). We then include only those cells in the design in which *at least one* VAR model has been sampled. This procedure returned 74 non-empty cells. We then sample those 74 cells until each of them contains 100 VAR models. We keep the cell size constant to render the results comparable across cells (see Supporting Information for a detailed description of this procedure).

This procedure returns a set of $74 \times 100 = 7400$ VAR models that includes essentially any stationary VAR model with $p = 6$ variables, and allows us to describe each model in the dimensions $O$ and $D$. For each of these VAR models, we generate 100 independent time series, each with $n = 500$ observations and with a burn-in period of $n_{\text{burn}} = 100$. We then estimate both the AR and the VAR model on the first $n = \{8, 9, \ldots, 499, 500\}$ observations of those time series. This yields a simulation study with $7400 \times 493$ (parameters $\times$ sample size) conditions, and for each of those conditions we have 100 replications. For each model, and each $n$, we compute the expected estimation error for both the AR model ($\text{EE}_{\text{AR}}$) and the VAR model ($\text{EE}_{\text{VAR}}$) model by averaging over the 100 replications. This means that while $\text{EE}_{\text{AR}}$ and $\text{EE}_{\text{VAR}}$ have different values depending on $n$ and the underlying model, we have averaged over the sampling variation.

## Simulation results

The simulation described above allows us to investigate the relative performance of AR and VAR models across different samples, sample sizes, and data-generating models. We define the estimation error as the mean squared error of the estimated parameters to the true parameters, and quantify the *relative* performance with two measures: the difference between the estimation errors of the AR and VAR models at a particular sample size, $\text{EE}_{\text{Diff}} = \text{EE}_{\text{AR}} - \text{EE}_{\text{VAR}}$; and, $n_e$, the sample size at which the VAR model outperforms the AR model ($\text{EE}_{\text{AR}} > \text{EE}_{\text{VAR}}$). In the following we examine the mean and variance of $\text{EE}_{\text{Diff}}$ and subsequently study $n_e$ and its dependence on the characteristics of the true VAR model.

Fig 1(a) shows the mean and standard deviation of $\text{EE}_{\text{Diff}}$ as a function of $n$, across all 7400 VAR models and 100 replications. The dashed line at $\text{EE}_{\text{Diff}} = 0$ indicates the point at which the estimation errors of the two models are equal. Below that line, the AR model performs better, that is, its parameter estimates are closer to the parameters of the true VAR model than the parameter estimates of the VAR model. We see that, across all models, we obtain a median $n_e$ = 89. Note that, out of all 740,000 simulated data sets, in only 23 cases the estimation error

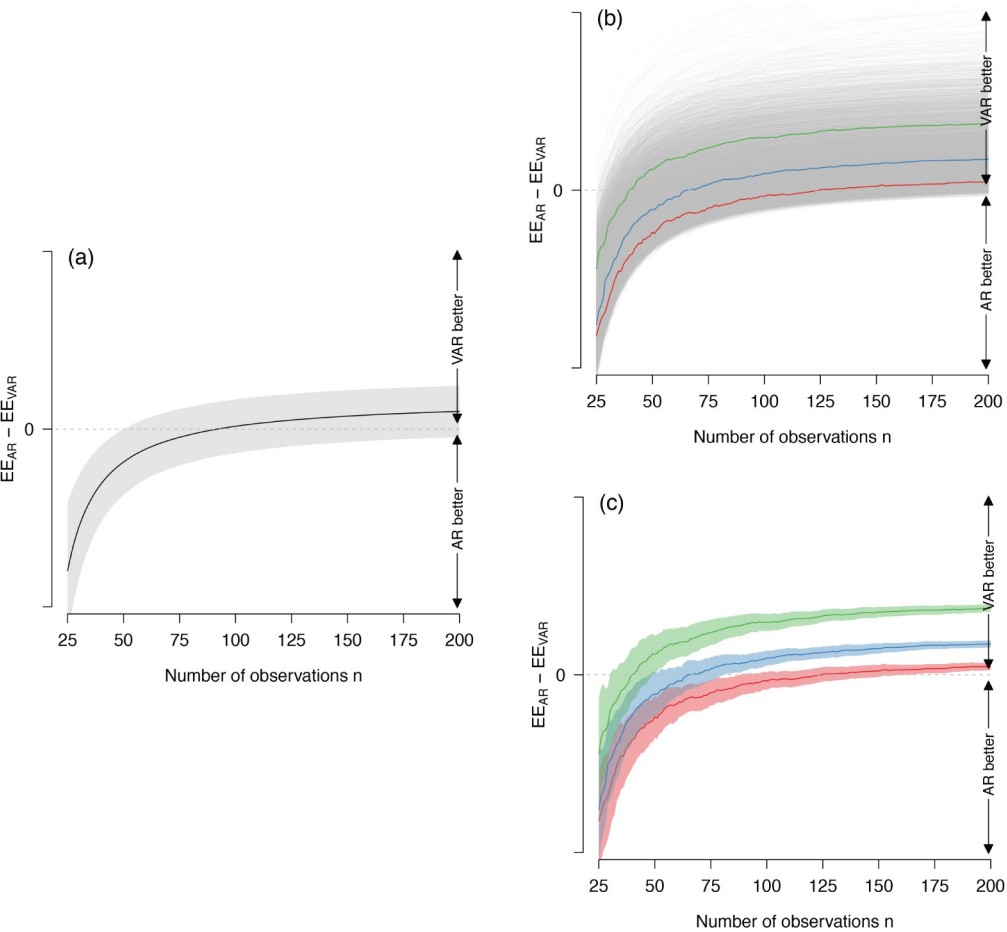

**Fig 1. Difference in estimation error of AR and VAR models ($EE_{Diff}$) across $n$ on three different levels of aggregation.** Panel (a) shows $EE_{Diff}$ averaged over replications and models, and the band shows the standard deviation over replications and models; panel (b) shows $EE_{Diff}$ for each model averaged across replications; and panel (c) shows the $EE_{Diff}$ averaged over replications for three specific models, and the bands show the standard deviation across 100 replications (sampling variation).

curves did not yet cross with an $n$ of 500. Notably, the variance around the difference in estimation error is substantial for all $n$. In the following we decompose this variance in variance due to sampling error, and variance due to differences in VAR matrices.

Panel (b) of Fig 1 displays the mean $EE_{Diff}$ for each of the 7400 VAR models, averaged across 100 replications. We see that the lines differ considerably and that $n_e$ substantially depends on the characteristics of the true VAR model. This shows that one cannot expect reliable recommendations with respect to $n_e$ that ignore the characteristics of the generating model. To illustrate the extent of the sampling variation of the models, we have chosen three particular VAR models (see coloured lines). Fig 1(c) shows that they exhibit considerable sampling variation. Note that, as the variance in (b) is due to differences in mean performance across VAR models, it does not decrease with $n$. In contrast, the variance in (c) depends on $n$ as it pertains to the sampling variance of a single VAR model, which decreases with the square root of the number of observations. While the mean $EE_{Diff}$ (shown in Fig 1(a)) gives a clear answer to the question of which $n$ is required for the VAR model to outperform the AR model

*on average*, both types of variation (see Fig 1(b) and 1(c)) show that for any *particular* VAR model it is difficult to determine which model performs better with the sample sizes typically available in psychological applications. However, we see that the sampling variation across replications is smaller than the variation across VAR models for most *n*. This means that if one has information about the parameters of the data-generating model, one can make much more precise statements about the sample size necessary for the VAR model to outperform the AR model.

The large degree of variation around EE$_{\text{Diff}}$ also highlights the potential pitfalls of generalizing the findings of Bulteel et al. [7] beyond the empirical data sets, which consist of 28, 52, and 95 individual time-series with an average number of 41, 70 and 70 time points, analyzed by the original authors. This is because (i) it is unlikely that their (in total) 175 time series appropriately represent the population of all plausible VAR matrices, (ii) their sample is subject to a substantial amount of sampling variation, and (iii) the absence of systematic variations of *n* does not allow a comprehensive answer to how relative performance relates to sample sizes.

Above we suggested that the relative performance of AR and VAR models (quantified by EE$_{\text{Diff}}$) depends on the characteristics *D* and *O* of the true VAR parameter matrix. In Fig 2(a) we show the median (across models in cells) *n* at which the estimation error of VAR becomes smaller than the estimation of AR (i.e., EE$_{\text{Diff}} > 0$), depending on the characteristics *D* and *O*. We see that the larger the average off-diagonal elements *O*, the lower the *n* at which VAR outperforms AR. This is what one would expect: when *O* is small (as indicated by the lowest rows of cells in Fig 2(a)), the true VAR model is actually very close to an AR model. In such a situation, the bias introduced by the AR model by setting the off-diagonal elements to zero leads to a relatively small estimation error. This trade-off between a simple model with high bias but low variance and a more complex model with low bias but high variance is well-known in the statistical literature as the *bias-variance trade-off* [8]. It therefore takes a considerable amount of observations until the variance of the VAR estimates becomes small enough for it to outperform the AR model. When *O* is large (indicated by the upper rows of cells), the bias of the AR model leads to a comparatively larger estimation error. Finally, we can also see that the size of the diagonal elements *D* is not as critical in determining $n_e$ as the size of the off-diagonal elements: Picking any row of cells in Fig 2(a), we can see that there is only a very small variation across columns, with larger *D* values appearing to lead to very slight decreases in $n_e$ in general. Note that the *O* characteristic also largely explains the vertical variation of the estimation error

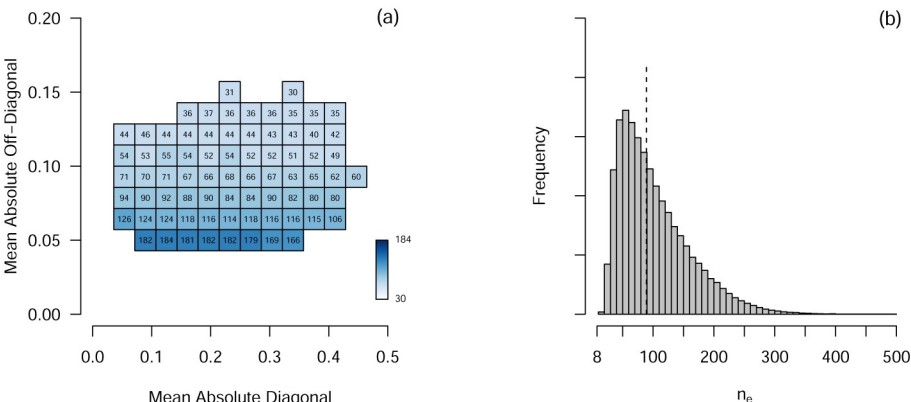

**Fig 2.** Left: $n_e$, the *n* at which estimation error becomes lower for the VAR than for the AR model, as a function of *D* and *O*. Right: Sampling distribution of $n_e$, the *n* at which the expected estimation error of the VAR model becomes lower than the expected estimation error of the AR model. The dashed line indicates the median of 89.

curves shown in Fig 1(b): the curves on top (small $n_e$) have high $O$, while the curves at the bottom (large $n_e$) have low $O$. Fig 2(b) collapses across these values and illustrates the sampling distribution of $n_e$, taking into account the likelihood of any particular VAR matrix (as specified by the mixed model estimated from the "MindMaastricht" data).

In summary, we used a simulation study to investigate the relative performance of AR and VAR models in a much larger space of plausible data-generating VAR models in psychological applications than considered by Bulteel et al. [7]. Next to investigating the *average* relative performance as a function of $n$, we also looked into the variation around averages. We showed that there is substantial variation both due to sampling error and differences in VAR matrices, which means that for a particular time series at hand it is difficult to select between AR and VAR with the $n$ available in typical psychological applications. Finally, we found that the size of the off-diagonal elements influences the relative performance of the VAR model more strongly than the size of the diagonal elements.

## Choosing between VAR and AR based on prediction error

In the previous section, we directly investigated the estimation errors of the AR and the VAR model in typical psychological applications and showed that the $n$ at which VAR becomes better than AR depends substantially on the characteristics of the true model. In practice, the true model is unknown, so we can neither look up the $n$ at which VAR outperforms AR in the above simulation study, nor can we compute the estimation error on the data at hand. Thus, to select between these models in practice, we may choose to use the prediction error which we can approximate using the data at hand, for instance by using a cross-validation scheme as suggested by Bulteel et al. [7]. However, since we are interested in estimation error, we require a link between prediction error and estimation error. In the remainder of this section we investigate this link and discuss the implications of this link for the model selection strategy suggested by Bulteel et al. [7], who use the "1 Standard Error Rule" (1SER) to select the model with lowest estimation error. Finally, we use our simulation study from above to directly compare the performance of the 1SER with model selection based only on the minimum prediction error.

### The relation between prediction error and estimation error

Bulteel et al. [7] suggest that the link between prediction error and estimation error is relatively straightforward: "[...] the number of observations $T$ [here $n$] that is needed for the VAR to become better than the AR is the same for the prediction MSE [mean squared error] as well as for the parameter accuracy [estimation error]" [7, p. 10]. More formally, this claim states that if $n_e$ is the number of observations at which the estimation errors of the AR and VAR model are equal, and if $n_p$ is the number of observation at which the prediction errors of the AR and VAR model are equal, and $n_{gap} = n_e - n_p$, then $n_{gap} = 0$. Bulteel et al. [7] do not specify the exact conditions under which this statement should hold, and elsewhere in the text suggest that this should be considered an approximate rather than an exact relationship. If this relationship were indeed approximate, it would still be interesting to study in which settings $n_{gap} > 0$ or $n_{gap} < 0$, as this bears on model selection, and so we will focus our investigation on quantifying $n_{gap}$ and investigating any potential systematic deviations from zero through simulation. Clearly, it would be unreasonable to expect that $n_{gap} = 0$ for *any* data set, since the observations in a given data set are subject to sampling error. We therefore interpret the statement of Bulteel et al. [7] as a statement about the *expectation* over errors of *any* given VAR model.

## Assessing $n_{gap}$ through simulation

We now use the results of the simulation study from the previous section to check whether indeed $n_{gap} = 0$ on average for all VAR models. To compute prediction error, we generate a test-set time series consisting of $n_{test} = 2000$ observations (using a burn-in period of $n_{burn} = 100$) for each of the 7400 VAR models described in the previous section. For each of the 100 replications of model and sample size condition, we average over the prediction errors which are obtained when estimated model parameters are evaluated on the test set. This is the out-of-sample prediction error (i.e., the expected generalization error) that Bulteel et al. [7] approximate with out-of-bag cross-validation error. We define prediction error as the mean squared error (MSE) of the predicted values relative to the true values in the test data set.

Fig 3 shows the estimation (solid lines) and prediction (dashed lines) errors for both the AR (black lines) and VAR (red lines) models as a function of $n$, averaged across the replications, for model A with $D = 0.068$ and $O = 0.092$ (left panel) and model B with $D = 0.337$ and $O = 0.051$ (right panel). For model A, we see that $n_{gap} < 0$, which shows that $n_{gap} = 0$ for all VAR models is incorrect. What consequences does this gap have for model selection? The negative gap implies that if the prediction errors for the AR and VAR model are the same, the VAR model should be selected, because its estimation error is smaller. In contrast, for model B we observe $n_{gap} > 0$. In this situation, if the prediction errors are equal, one should select the AR model because it incurs smaller estimation error. Clearly, $n_{gap}$ differs between the two models, and this difference matters for model selection.

So far we only investigated $n_{gap}$ for two individual VAR models. Fig 4(a) shows the distribution of the expected $n_{gap}$ across all VAR models, computed by averaging over 100 replications. Note that for 31 out of 7400 models the curves of prediction errors and estimation errors did not cross within $n \in \{8, 9, \ldots, 499, 500\}$. The results in Fig 4 are therefore computed on 7369 models.

Each of the data points in the histogram in Fig 4(a) corresponds to the *expected* $n_{gap}$ of one of the 7369 models. We see that the expected $n_{gap}$ has a right skewed distribution with a mode at zero. This allows us to make a precise statement regarding the crossing of estimation and prediction errors described above: while the most common value of $n_{gap}$ is zero, most expected $n_{gap}$ are not zero. In fact, $n_{gap}$ shows substantial variation across different VAR models. Explaining the variance of $n_{gap}$ is interesting, because $n_{gap}$ has direct consequences for model

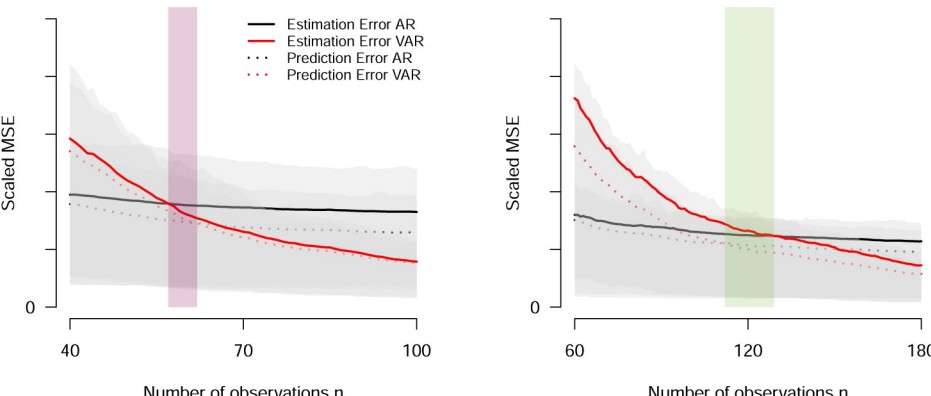

**Fig 3.** Scaled Mean Squared Error (MSE) of estimation (solid lines) and prediction errors (dashed lines) for both the AR (black lines) and VAR (red lines) models as a function of $n$, separately for model A with $D = 0.068$ and $O = 0.092$ (left panel) and model B with $D = 0.337$ and $O = 0.051$ (right panel). The red and green shaded area indicates the median $n_{gap}$, and the grey shaded area shows the 20% and 80% quantiles across the 100 replications per model.

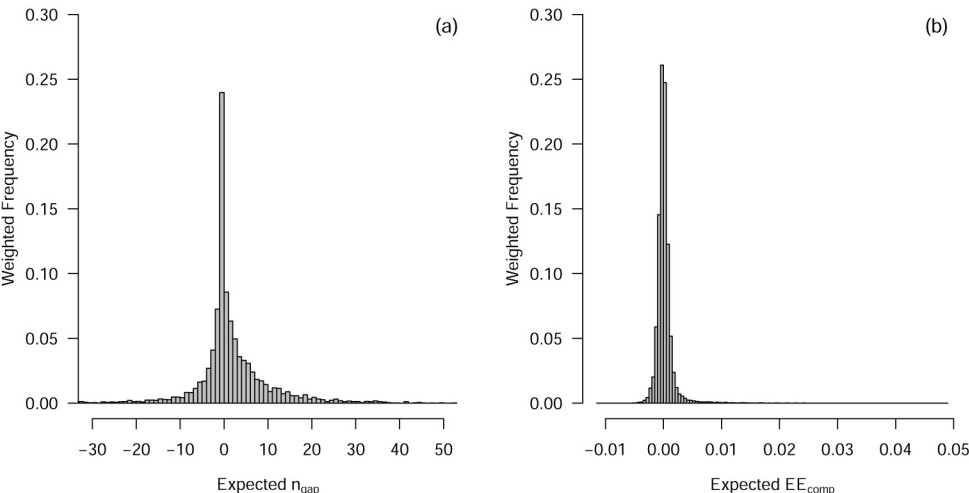

**Fig 4.** Panel (a) displays the distribution of the expected $n_{\text{gap}}$ across all 7369 VAR models, computed by averaging over 100 replications, and weighted by the probability defined by the original mixed model. Panel (b) shows the distribution of *non-zero* $\text{EE}_{\text{comp}}$ across all $n$, 7369 VAR models, averaged across replications and weighted by the probability defined by the original mixed model.

selection. If we can relate the $n_{\text{gap}}$ to characteristics of the $\Phi$ matrix, it is possible to make more specific statements with respect to when to apply a bias towards the AR or VAR model, when the prediction errors are the same or very similar. Note that such a function from $\Phi$ to $n_{\text{gap}}$ must exist, because the only way the 7400 models differ is in their entries of the VAR parameter matrix $\Phi$. However, this function may be very complicated. For example, the Pearson correlation of $n_{\text{gap}}$ with $D$ and $O$ are 0.21 and −0.02, respectively. Predicting $n_{\text{gap}}$ by $D$ and $O$ including the interaction term with linear regression achieves $R^2 = 0.048$. This shows that a simple linear model including $D$ and $O$ is not sufficient to describe the relationship between $n_{\text{gap}}$ and $\Phi$. Future research could look into better approximations of this relationship. If successful, one could build new model selection strategies on reliable predictions of $n_{\text{gap}}$ from empirical data.

## Performance of the "1 Standard Error Rule"

Bulteel et al. [7] propose, in the words of Hastie et al., to "[...] choose the most parsimonious model whose error is no more than one standard error above the error of the best model." [8], p. 244]. This model selection criteria is known as the "1 Standard Error Rule" (1SER) and is suggested by Hastie and colleagues as a method of choosing a model with the minimal out-of-sample prediction error (which is typically unknown), on the basis of out-of-bag prediction error (acquired with cross-validation techniques).

Making inferences from prediction error to estimation error requires a link between the two. Bulteel et al. [7] provide this link by suggesting that $n_{\text{gap}} = 0$ (or $n_{\text{gap}} \approx 0$). However, they do not provide justification for why the 1SER should outperform simply selecting the model with the lowest prediction error. Above we showed that $n_{\text{gap}} = 0$ does not hold for all VAR models. In fact, it is this result that explains why the 1SER can perform better than selecting the model with the lowest prediction error. Specifically, this is the case when $n_{\text{gap}} > 0$, which characterizes the situation that the prediction error for VAR is lower than for AR while at the same time the estimation error of VAR is higher than for AR. In such a situation, a bias

towards the AR model can be favorable. In contrast, if $n_{\text{gap}} < 0$ and the prediction error of AR is lower than for VAR, even though the estimation error of VAR is lower than for AR, such a bias would be unfavorable. In the following, we assess the relative performance of the 1SER and simply selecting the model with lowest prediction error, both on average and as a function of $n$.

In order to quantify the relative performance of both model selection strategies, we take the prediction and estimation errors of the 7400 VAR models estimated on $n \in \{8, 9, \ldots, 499, 500\}$ and for each model, and each $n$, select between the AR and VAR model in two different ways: (1) by simply selecting the model with the lowest prediction error, and (2) by applying the 1SER (using the standard-deviation of the out-of-sample prediction error across 100 training sets). For each of the two strategies, we then subtract the estimation error of the selected model ($EE_{\text{sel}}$) from the estimation error of the model with the lowest estimation error ($EE_{\text{best}}$). The difference $EE_{\text{diff}} = EE_{\text{best}} - EE_{\text{sel}}$ equals zero if the model with lower estimation error has been selected, and is negative if the model with higher estimation error has been selected. Subsequently, we compute

$$EE_{\text{comp}} = EE_{\text{diff}}^{(2)} - EE_{\text{diff}}^{(1)} \ ,$$

where $EE_{\text{diff}}^{(2)}$ is the difference obtained using (2), and $EE_{\text{diff}}^{(1)}$ is the difference obtained using (1). The resulting value of $EE_{\text{comp}}$ allows us to compare the performance of the two model selection strategies. That is, if $EE_{\text{comp}} < 0$, simply selecting the model with lowest prediction error performs better, and if $EE_{\text{comp}} > 0$, the 1SER performs better.

Fig 4(b) shows the distribution of *non-zero* $EE_{\text{comp}}$ across all 7400 VAR models, averaged over replications, and weighted by the probability given by the original mixed model. The only interesting cases when comparing model selection procedures are the cases in which they disagree. Therefore, we analyze only those cases for which $EE_{\text{comp}} \neq 0$. Note that for all but 2 of the 7400 models there is some $n$ at which the two decision rules in question choose a different model. We find that using the 1SER is better in 50.1% of cases (where each case is weighted by the probability of the corresponding model). This would suggest that it makes essentially no difference whether we use the 1SER or select the model with lowest prediction error. However, these proportions average over the number of observations $n$ and therefore cannot reveal differences in relative performance for different sample sizes.

Fig 5(a) shows $EE_{\text{comp}}$ as a function of $n$, averaged across all 7400 models. Because the VAR prediction error is huge for very small $n$, both model selection strategies choose the same model, resulting in $EE_{\text{comp}} = 0$ for those $n$. However, from around $n = 10$ on until around $n = 60$, $EE_{\text{comp}}$ is substantially positive, indicating that the 1SER outperforms simply selecting the model with the lowest prediction error by a large margin. However, for $n > 60$ we see that $EE_{\text{comp}}$ approaches zero and then becomes slightly negative. The latter is also illustrated in panel (b), which displays the weighted proportion of models in which the 1SER is better (i.e., $EE_{\text{comp}} > 0$). The explanation of this curve has three parts. First, $n_{\text{gap}}$ tends to be *larger* if the gap is located at a *small n* (Pearson correlation $r = -0.15$). If $n_{\text{gap}}$ is large (and therefore positive), the AR model has lower estimation error than the VAR model, even though the prediction errors are the same (compare Fig 5(b)). In such situations, biasing model selection towards selecting the AR model is advantageous. Since the 1SER constitutes a bias towards the AR model, it performs better for small $n$. Second, this also explains why the 1SER performs worse than simply selecting the model with lowest prediction error for large $n$: here the gap is small (negative), indicating that if the prediction errors are the same, the VAR model performs better. Clearly, in such a situation, providing a bias towards AR is disadvantageous. Therefore, the 1SER performs worse. Finally, why does the curve get closer and closer to zero? The reason

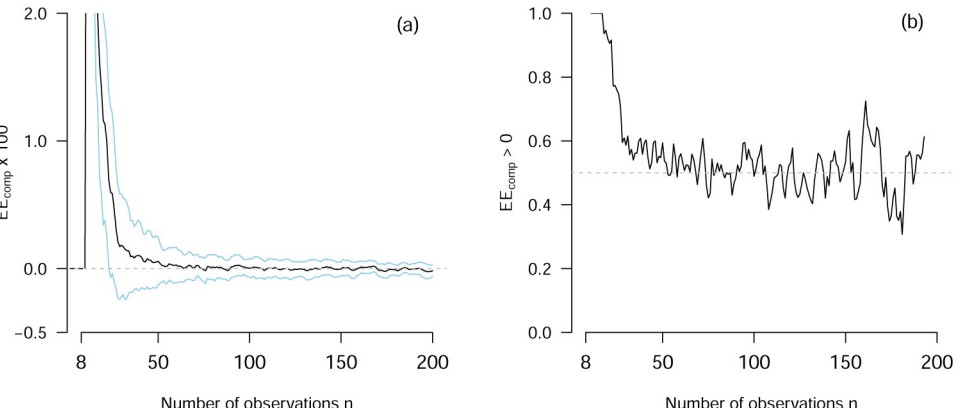

**Fig 5.** Panel (a) displays $EE_{comp}$ averaged across 7400 models as a function of $n$ (black line) and the standard deviation around the average (blue line). Panel (b) displays, for each $n$, the proportion of times that $EE_{comp} > 0$ across 7400 models (i.e., the proportion of 1SER performing better).

is that the standard error converges to zero with (the square root of) the number of observations, and therefore the probability that both rules select the same model approaches 1 as $n$ goes to infinity.

To summarize, we found that the 1SER is better than simply selecting the model with the lowest prediction error only in 50.1% of the cases in which the two rules do not select the same model. However, when looking at the relative performance as a function of $n$, we found that the 1SER is better than selecting the model with lowest prediction error until around $n = 60$, and worse above. Finally, we were able to explain the dependence of the relative performance on $n$ with the fact that $n_{gap}$ is larger when it occurs at a smaller $n$. For applied researchers these results suggest that, for VAR models with $p = 6$ variables, the 1SER should be applied for $n < 60$.

## Discussion

In this paper we provided an extended analysis of the problem studied by Bulteel et al. [7] by using a simulation study to (a) map out the relative performance of AR and VAR models in typical psychological applications as a function of the number of observations $n$, and (b) investigate how to choose between AR and VAR models in practice. We found that, averaged over all models considered in our simulation, the VAR model outperforms the AR model for $n > 89$ observations in terms of estimation error. In addition, we show that and explain why the 1SE rule proposed by Bulteel et al. [7] performs better than selecting the model with the lowest prediction error when $n$ is small.

Next to the *average* estimation errors of AR and VAR models, we also investigated the *variance* around those averages. We decomposed this variance in variance due to different true VAR models, and variance due to sampling. The variance across different VAR models showed that the relative performance, that is, the $n$ at which VAR becomes better than AR ($n_e$) depends on the characteristics of the true VAR parameter matrix $\Phi$. For example, if the true VAR model is very close to an AR model, it takes more observations until the VAR model outperforms the AR model. This shows that one cannot expect reliable recommendations with respect to $n_e$ that ignore the characteristics of the generating model: $n_e$ critically depends on the size of the off-diagonal elements present in the data-generating model. The size of the

sampling variation also indicates that, for many of the considered sample sizes, whether the VAR or AR model will have lower estimation error largely depends on the specific sample at hand. This implies that it is difficult to select the model with lowest estimation error with the sample sizes available in typical psychological applications.

The second question we investigated was: how should one choose between the AR and VAR model for a given data set? Bulteel et al. [7] suggest that, for any VAR model, the $n$ at which the prediction errors of both models are equal is, in expectation, (approximately) the same $n$ at which their estimation errors are equal (i.e., $n_{gap} \approx 0$). Combining this claim with a preference towards the more parsimonious AR model, they proposed using the "1 Standard Error Rule", according to which one should select the AR model if its prediction error is not more than one standard error above the prediction error of the VAR model, and choose the model with lowest prediction error otherwise. We showed that the expected $n_{gap}$ varies as a function of the parameter matrix of the true VAR model. Using the relationship between estimation and prediction error we were able to explain when the 1SER is expected to perform better than selecting the model with lowest prediction error. In addition, we showed via simulation that the 1SER performs better than selecting the model with the lowest prediction error for $n < 60$, in cases where those decision rules select conflicting models. Our simulations also showed that as $n \to \infty$, both decision rules converge to selecting the same model. This means that there is a relatively small range of sample sizes in which these decision rules lead to contradictory model selections for a given data-generating system. We recommend that researcher wishing to use prediction error to choose between these models examine both the 1SER and lowest prediction error rules, and in cases of conflict between the two, use the 1SER for low ($n < 60$) sample sizes.

The relative performance of the AR and VAR model shown in our simulations can be understood in terms of the bias-variance trade-off. Because the AR model sets all off-diagonal elements to zero, it has a bias that is constant and independent of $n$. In contrast, the VAR model has a bias of zero, since the true model is a VAR model. This is why a VAR model will always perform better than (or at least as good as, if the all off-diagonal elements of the true VAR model are zero) an AR model as $n \to \infty$. However, for finite sample sizes the variance of the estimates of the two models are different: while both variances converge to zero as $n \to \infty$, for finite samples the variance of VAR parameters is much larger than the variance of AR parameters, especially for small $n$. This allows for the situation that the biased simpler model is showing a smaller error, even though the true model is in the class of the more complex model. This trade-off between bias and variance also explains the relative performance of AR and VAR models: From Fig 3 we saw that for small $n$, the variance of the VAR estimates is so large that the error is larger than the error of the AR model, despite the bias of the AR model. However, with increasing $n$, the variance of the estimates of both models approaches zero. This means that the larger $n$, the more the bias of the AR model contributes to its error. Thus, at some $n$ the error of the VAR model becomes smaller than the error of the AR model. We agree with Bulteel et al. [7] that the fact that a simple (and possibly implausible) model can outperform a complex (and more plausible) model, even though the true model is in the class of the more complex model, is underappreciated in the psychological literature.

An interesting question we did not discuss in our paper is: which model should we choose if the AR and VAR models have equal estimation error? Since we defined the quality of a model by its estimation error, we could simply pick one of the two models at random. However, their model parameters are likely to be very different. The estimation error of the AR model comes mostly from setting off-diagonal elements incorrectly to zero, while the estimation error of the VAR model comes mostly from incorrectly estimating off-diagonal elements. In terms of the types of errors produced by the two models, the AR model will almost

exclusively produce false negatives, while the VAR model will produce almost exclusively false positives. A specification of the cost of false positives/negatives in a given analysis may allow to choose between models when the estimation errors are the same or very similar. For example, in an exploratory analysis one might accept more false positives in order to avoid false negatives.

Throughout the paper we compared the AR model to the VAR model. However, we believe that it is unnecessarily restrictive to choose only between those extremes (all off-diagonal elements zero vs. all off-diagonal elements nonzero). The AR model, by imposing independence between processes, presents a theoretically implausible model for many psychological processes. Applied researchers who estimate the VAR model may be primarily interested in the recovery of cross-lagged effects rather than auto-regressive parameters, for example to determine which processes are dependent on one another (as evidenced by frequent discussions of Granger causality [11] In such settings, one could estimate VAR models with a constraint that limits the number of nonzero parameters or penalizes their size [12, 13]. This would allow the recovery of large off-diagonal elements without the high variance of estimates in the standard VAR model. Similarly, one could estimate a VAR model and, instead of comparing it to an AR model and thus testing the nullity of the off-diagonal elements jointly, test the nullity of the off-diagonal elements of the VAR matrix individually. Further investigation of these alternatives in future studies would provide a more complete picture to applied researchers.

It is important to keep the following limitations of our simulation study in mind. First, we claimed that the 7400 models we sampled from the mixed model obtained from the "Mind-Maastricht" data represent typical applications in psychology. One could argue that there are sets of VAR models that are plausible in psychological applications that are not included in our set of models. While this is a theoretical possibility, we consider this extremely unlikely, since we heavily sampled the mixed model stratified by $O$ and $D$. Any VAR model that is not similar to a model in our set of considered VAR models is therefore most likely non-stationary. When presenting our results we weighted all models by their frequency given the estimated mixed model in order to avoid giving too much weight to unusual VAR models. This means that it could be that the weighting obtained from the mixed model does not represent the frequency of VAR models in psychological applications well. While we consider this unlikely, we also used a uniform weighting across VAR models as a robustness check which left all main conclusions unchanged. A second limitation is that we only considered VAR models with $p = 6$ variables. While this is not a shortcoming compared to Bulteel et al. [7] who use VAR models with 6, 6, and 8 variables, the results shown in the present paper would likely change when considering more or less than six variables. Specifically, we expect that the $n$ at which VAR outperforms AR becomes larger when more variables are included in the model, and smaller when less variables are included. This change may be nonlinear in nature: As we add variables to the model, we would expect the variance of the VAR model to grow much quicker than the variance of the AR model, since in the former case we need to estimate $p^2$ parameters, and in the latter only $p$. However, the bias of the AR model also grows with each new variable added, with $p^2 - p$ elements set to zero in each case, and so again, this will largely depend on the data-generating system at hand. Similarly, we would expect that for models with more variables the 1SER outperforms selecting the model with lowest prediction error for sample sizes larger than 60. While the exact values will change for larger $p$, we expect that the general relationships between $n$, $O$, and $D$ extend to any number of variables $p$.

Although Bulteel et al. [7] also consider mixed VAR and AR models, in the simulation studies presented above we focus exclusively on single-subject time-series for simplicity. Mixed models can be seen as a form of regularization, in which individual parameter estimates are shrunk towards the group-level mean if the number of observations $n$ is small. One would

expect that for small $n$, the use of mixed models would improve the estimation and prediction errors of both models, which is also what Bulteel et al. [7] report in their results. Indeed, mixed models are expected to improve the performance of VAR methods relative to AR, and thus may be a solution to the relatively poor performance of the VAR model we observe in sample sizes realistic for psychological applications. The reason is that the differential performance of AR and VAR models can be understood in terms of a bias-variance trade-off, where AR models are biased but have lower variance than VAR methods. The use of mixed VAR models should decrease this variance through shrinkage in small $n$ settings [14, 15]. The precise effect of using mixed models depends on the variance of parameters across individuals; however, we do not expect the general pattern of results reported here to change when moving from single-subject to mixed settings.

Future research could extend the analysis shown here to VAR models with less than or greater than six variables, which would allow to generalize the simulation results to more situations encountered in psychological applications. Another interesting avenue for future research would be to investigate the link between $n_{gap}$ and the VAR parameter matrix $\Phi$. Since $n_{gap}$ has direct implications for model selection, such a link could possibly be used to construct improved model selection procedures. It would be useful to extend the simulation study in this paper to constrained estimation such as the LASSO, especially since those methods are already applied in practice [16]. Finally, it would be useful to study the performance of mixed VAR models in a simulation setting, and perhaps compare this approach to alternative methods of using group-level information in individual time-series analysis, such as GIMME, an approach originally developed for the analysis of neuroimaging data [17]. Early simulation studies have assessed the performance of mixed AR models in recovering fixed effects using Bayesian estimation techniques [18], but these analyses have yet to be extended to mixed VAR models or the recovery of individual-specific random effects.

To sum up, we used simulations to study the relative performance of AR and VAR models in settings typical for psychological applications. We showed that, on average, we need sample sizes approaching $n = 89$ for single-subject VAR models to outperform AR models. While this may seem like a relatively large sample size requirement, such longer time series are becoming more common in psychological research [19, 20] Decomposing this variance showed that (i) one cannot expect reliable statements with respect to the relative performance of the AR and VAR models that ignore the characteristics of the generating model, and (ii) that choosing reliably between AR and VAR models is difficult for most sample sizes typically available in psychological research. Finally, we provided a theoretical explanation for when the "1 Standard Error Rule" outperforms simply selecting the model with lowest prediction error, and showed that the 1SER performs better when $n$ is small.

## Supporting information

**S1 Fig. *D* and *O* values for the initially sampled 10000 VAR models.**
(PDF)

## Acknowledgments

We would like to thank Don van den Bergh, Riet van Bork, Denny Borsboom, Max Hinne, Lourens Waldorp, and two anonymous reviewers for their helpful comments on earlier versions of this paper.

## Author Contributions

**Conceptualization:** Fabian Dablander, Oisín Ryan, Jonas M. B. Haslbeck.

**Formal analysis:** Fabian Dablander, Oisín Ryan, Jonas M. B. Haslbeck.

**Writing – original draft:** Fabian Dablander, Oisín Ryan, Jonas M. B. Haslbeck.

**Writing – review & editing:** Fabian Dablander, Oisín Ryan, Jonas M. B. Haslbeck.

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
