## [Decision Letter · Decision Letter 0]

14 Apr 2020

PONE-D-20-03592

Choosing between AR(1) and VAR(1) Models in Typical Psychological Applications

PLOS ONE

Dear Mr Haslbeck,

Thank you for submitting your manuscript to PLOS ONE. After careful consideration, we feel that it has merit but does not fully meet PLOS ONE’s publication criteria as it currently stands. Therefore, we invite you to submit a revised version of the manuscript that addresses the points raised during the review process.

Please, take into account all the considerations raised by the reviewers.

We would appreciate receiving your revised manuscript by May 22 2020 11:59PM. To enhance the reproducibility of your results, we recommend that if applicable you deposit your laboratory protocols in protocols.io, where a protocol can be assigned its own identifier (DOI) such that it can be cited independently in the future. For instructions see: http://journals.plos.org/plosone/s/submission-guidelines#loc-laboratory-protocols

We look forward to receiving your revised manuscript.

Kind regards,

Miguel Angel Sánchez Granero

Academic Editor

PLOS ONE

Journal Requirements:

Reviewers' comments:

Reviewer's Responses to Questions

**Comments to the Author**

1. Is the manuscript technically sound, and do the data support the conclusions?

Reviewer #1: Yes

Reviewer #2: Partly

2. Has the statistical analysis been performed appropriately and rigorously? 

Reviewer #1: Yes

Reviewer #2: Yes

3. Have the authors made all data underlying the findings in their manuscript fully available?

Reviewer #1: Yes

Reviewer #2: Yes

4. Is the manuscript presented in an intelligible fashion and written in standard English?

Reviewer #1: Yes

Reviewer #2: Yes

5. Review Comments to the Author

Reviewer #1: The authors conducted comprehensive simulations to examine the performance of AR vs. VAR model using typical psychological time series data, and compared their performance in estimation error and prediction error as related to the length of time series and the characteristics of the true model. The study extended from Bulteel et al. (2018) and its results make a major contribution to the literature. In general, the manuscript is well written and clearly organized. I have a few comments and suggestions for the authors to consider when revising their manuscript.

My first general concern is on the “typical” part of the study. The authors, and the Bulteel et al. (2018) as well, fail to elaborate the main reason for the application of VAR model. More than often for applied researchers, they choose to use the VAR model because they are interested to know whether one variable A is related to another variable B at a later time (i.e., cross-lagged paths), after controlling for B at previous time. In other words, one is interested to know whether A has added value in terms of the prediction of B, and the choice of A and B are theoretically derived. From this point of view, it is theoretically meaningful to adopt VAR model rather than AR model. In such situation, the research question becomes whether VAR can accurately recover the cross-lagged links between variables, rather than whether AR outperforms VAR, under some conditions.

Relatedly, the authors initially claimed that the length of most applied psychological time series data fall between 30 to 200. It is important to note that the MindMaastricht dataset, where the current simulations are based on, in my mind are not typically psychological time series data (52 individuals with an average of 41 measurements on 6 variables). All three data used by Bulteel et al. (2018) face the same issue as well (individuals fewer than 100, lengths between 41 and 70). From my reading of the applied literature, most studies tend to have a lot more participants with shorter time series and fewer variables (at least those examined in the VAR model). Whether the mean number of 92 based on estimation error, or the number of 60 for prediction performance, they are all beyond the length of most typical psychological time series data. Does it mean that applied researchers should just always go with the AR model? The authors should discuss this point.

The authors encouraged future studies with more than 6 variables. However, with fewer than 6 variables considered, how would the current findings hold (I reckon n for both estimation and prediction errors likely will go down)? It is likely that it may take fewer n for VAR to outperform AR.

For each VAR model (R and D) condition, 100 independent time series were simulated. These are more referred to as “replications” for each model design condition, rather than “iterations” (e.g., page 5 line 148). The authors should revise the term where applicable throughout the manuscript.

The authors simulated n = 500 for estimation simulation but n = 2000 for prediction simulation. From the results and discussions, it appears that 2000 does not matter too much. Discussions are needed regarding this point.

Figure 4b and on page 11 line 350, the authors should state how many cases have EEcomp unequal to zero.

The authors mentioned mixed models – some recent simulation work on DSEM should be cited, which have shown satisfying estimation results for VAR. Furthermore, the authors should briefly discuss the subgroup/mixture approach when there are distinct subgroups of time series patterns (e.g., GIMME).

Minor comments

When referring to the mixed effects examined in Bulteel et al. (2018), at least for the first time (page 2 line 40), it would be helpful to clarify it refers to multilevel model with random effects.

On page 4 line 123, it should be Figure 6 in the supplementary materials.

On page 7 line 201, two “have”s; line 202, two “the”s.

Reviewer #2: The authors present results from a series of simulation studies examining the performance of AR and VAR models. Results assist the reader in determining which model structure (i.e., AR versus VAR) to use when modeling n=1 time series data. I appreciate and admire the clarity with which the authors describe complex methodology and present their results. I believe that this paper will be a valuable contribution to the field of psychological time series. Below I have outlined suggestions to facilitate the connection of the theoretical nature of this manuscript to applied psychological data.

1) Page 3 and 4: I appreciate the novel methods the authors used to generate their simulated data through the use of parameters, R & D. However, I am concerned that this method introduces artifacts into the sampling scheme, due to the fact that there is a correlation between R & D (as shown in Figure 6). Thus, it seems that there would be bias in the models generated with this technique. In general, although the authors provide some justification for using R & D, it would be helpful for the author to provide further explanation of their parameterization methods in light of this correlation. In particular, it seems that this correlation may be artificially induced by the authors’ definition of R & D. For example, a theorem from linear algebra states that the sum of the eigenvalues of a matrix (i.e., D) is equal to the sum of its diagonal elements (i.e., it’s trace, in this case the AR parameters included in the numerator of R). Hence, the numerator of R is essentially D. This suggests that the R-D parameterization is likely responsible for the correlation in the simulation samples. I recommend that the authors acknowledge this in their description of their parameterization methods. Additionally, I recommend that they examine the correlation between R & D to demonstrate that this correlation is sufficiently low so as to not overly bias the simulation data. Finally, I strongly suggest that authors reformulate R so that it is free from the influences of this correlation, such as by using the current denominator of R. This would allow for the modeling of autoregressive effects (i.e., D) and cross-lagged effects (i.e., denominator of R), independently.

2) I think it may be useful for the authors to provide more recommendations for the design of psychological time series studies based on their data. In other words, are there suggestions for how applied researchers should implement these findings?

3a) For example, do these results support the recommendation of collecting more observations in general?

3b) Lines 443-455 refer to several theoretical points about choosing between VAR and AR models under the condition of equal estimation error. Given that applied researchers may want to select one model over the other for hypothesis-testing reasons (e.g., testing the AR effect of mood versus including the cross-lagged effect of anxiety on mood), could you provide clarification on whether an applied researcher would be able to test for estimation error equivalence using empirical data? If that is not possible, I believe it may be helpful to state this explicitly.

3c) Line 385: In regards to comparing the 1SER rule versus selecting the model with the lower prediction error, what should applied researchers take away from these results if they are working with data with n > 60?

4) Line 173: Could you clarify what is meant by specifying the data generating model and how a researcher would do this using empirical data?

5) Line 509: I recommend rephrasing this sentence to specify that the relative performance of AR and VAR models were studied using simulations of data generated from typical psychological applications.

6) Line 24 = missing the word, “the”?

Overall, I appreciate the authors’ contribution the field of time series psychometrics. I hope that the authors find my comments helpful in assisting them with revising the draft for publication.

6. PLOS authors have the option to publish the peer review history of their article (what does this mean?). If published, this will include your full peer review and any attached files.

Reviewer #1: Yes: Yao Zheng

Reviewer #2: No

---

## [Author Response · Author response to Decision Letter 0]

28 Jul 2020

We have uploaded a file that responds in detail to the reviewer and editor comments. However, we have pasted them here as well:

Dear Editor,

Thank you for sending the comments of the reviewer and the Associate Editor. The comments and the close reading especially of the Associate Editor helped us again to make important improvements to our manuscript.

We append our responses to the reviewer's and Associate Editor's comments at the end of this letter.

Note that based on the comments of Reviewer 2, we re-ran the main simulation part of our study, using a slightly different sampling scheme, based on the size of the off-diagonal elements and diagonal elements ($O$ and $D$) respectively. This was largely done to aid the interpretation of our results as depicted in Figure 2. This new simulation has not changed our main results in any way, and we note it here mainly to draw attention to slight numerical differences that appear in the new manuscript. For instance, the median sample size requirement of $n_e = 92$ discussed by Reviewer 1 has decreased slightly to $n_e = 89$ in the new manuscript. A full discussion of these changes is given in our reply to Reviewer 2's comments.

\\closing{Kind regards,}

\\newpage 

\\Large

\\textbf{Reviewer 1}

\\normalsize

\\textbf{Comment 1}

\\begin{displayquote}

My first general concern is on the “typical” part of the study. The authors, and the Bulteel et al. (2018) as well, fail to elaborate the main reason for the application of VAR model. More than often for applied researchers, they choose to use the VAR model because they are interested to know whether one variable A is related to another variable B at a later time (i.e., cross-lagged paths), after controlling for B at previous time. In other words, one is interested to know whether A has added value in terms of the prediction of B, and the choice of A and B are theoretically derived. From this point of view, it is theoretically meaningful to adopt VAR model rather than AR model. In such situation, the research question becomes whether VAR can accurately recover the cross-lagged links between variables, rather than whether AR outperforms VAR, under some conditions.

\\end{displayquote}

We agree with the reviewer on this point, and have added a clarification on the theoretical choice of VAR over AR models in the discussion (lines 447 - 452, below). As a follow-up to Bulteel et al., a full investigation of cross-lagged parameter recovery was beyond the scope of the current paper. However, sample size requirements for when the VAR outperforms the AR model in estimation error is likely to be a lower bound on the sample size requirement for accurate recovery of cross-lagged parameters, as it indicates at what sample size the cross-lagged parameter estimation performs better in approximating the true parameter set than guessing zero for all cross-lagged parameters.

Added text:

“Throughout the paper we compared the AR model to the VAR model. However, we believe that it is unnecessarily restricting to choose only between those extremes (all off-diagonal elements zero vs. all off-diagonal elements nonzero).The AR model, by imposing independence between processes, presents a theoretically implausible model for many psychological processes. Applied researchers who estimate the VAR model may be primarily interested in the recovery of cross-lagged effects rather than auto-regressive parameters, for example to determine which processes are dependent on one another (as evidenced by frequent discussions of Granger causality [11] in these settings). In such settings, one could estimate VAR models with a constraint that limits the number of nonzero parameters or penalizes their size [12,13]. This would allow the recovery of large off-diagonal elements without the high variance of estimates in the standard VAR model. Similarly, one could estimate a VAR model and, instead of comparing it to an AR model and thus testing the nullity of the off-diagonal elements jointly, test the nullity of the off-diagonal elements of the VAR matrix individually. Further investigation of these alternatives would provide a more complete picture to applied researchers in future studies.”

\\pagebreak

\\textbf{Comment 2}

\\begin{displayquote}

 Relatedly, the authors initially claimed that the length of most applied psychological time series data fall between 30 to 200. It is important to note that the MindMaastricht dataset, where the current simulations are based on, in my mind are not typically psychological time series data (52 individuals with an average of 41 measurements on 6 variables). All three data used by Bulteel et al. (2018) face the same issue as well (individuals fewer than 100, lengths between 41 and 70). From my reading of the applied literature, most studies tend to have a lot more participants with shorter time series and fewer variables (at least those examined in the VAR model). Whether the mean number of 92 based on estimation error, or the number of 60 for prediction performance, they are all beyond the length of most typical psychological time series data. Does it mean that applied researchers should just always go with the AR model? The authors should discuss this point.

\\end{displayquote}

We agree with the reviewer that sample size requirements of 89/92 or so repeated measurements may be a tall order in many settings, that many studies utilize multiple-subjects rather than single-subject designs. 

With regards to the sample size requirements, it is important to note an additional finding of our study: that although the average sample size requirement for VAR to outperform AR is 89, there is a very large degree of variation around this value. This variation is largely determined by the absolute size of the off-diagonal elements: From Figure 2 we can now see clearly that data-generating mechanisms with higher off-diagonal elements may require as little as half that many observations. We have emphasised this more clearly in the discussion section (lines 392 - 395):

“This shows that one cannot expect reliable recommendations with respect to $n_\\text{e}$ that ignore the characteristics of the generating model: $n_e$ critically depends on the size of the off-diagonal elements present in the data-generating model.”

While the reviewer states that even sample sizes of 40 observations per person appear unrealistic to them, it should be noted that more and more psychological studies are collecting longer and longer time-series, particularly in the domain of clinical psychology. Two recent examples include Wichers et al. (2016), who collected a single subject time series dataset of 1478 repeated measurements and Helmich et al (2020) which used data consisting of 100 repeated measurements each of 329 individuals. Finally, the availability of relatively long multiple-subjects data also opens up the possibility of using mixed effects / multilevel models. The use of these models will certainly decrease the number of measurements needed per person needed to recover model parameters, and although the study of those models was beyond the scope of the current paper, we believe that this is an important topic for future research. In order to reflect this we have added extra detail to this in the discussion section, on lines 504 - 507:

“Indeed, mixed models are expected to improve the performance of VAR methods relative to AR, and thus may be a solution to the relatively poor performance of the VAR model we observe in sample sizes realistic for psychological applications.”

And in addition on lines 529 - 542:

“To sum up, we studied the relative performance of AR and VAR models in simulations of typical psychological applications. We were able to make clear statements about the average performance of VAR models, which showed that, on average, we need sample sizes approaching $n = 89$ for single-subject VAR models to outperform AR models. While this may seem like a relatively large sample size requirement, such longer time series are becoming more common in psychological research \\cite{wichers2016critical, helmich2020sudden} and mixed models may allow for acceptable performance for shorter time series, though much research on that topic is still required. Importantly, we also found the variance around this average sample size to be considerable, with the variation largely a function of the average absolute value of the off-diagonal (i.e. cross-lagged) effects. Decomposing this variance showed that (i) one cannot expect reliable statements with respect to the relative performance of the AR and VAR models that ignore the characteristics of the generating model, and (ii) that choosing reliably between AR and VAR models is difficult for most sample sizes typically available in psychological research.”

Finally, we do not agree with the reviewer that our results suggest that researchers should necessarily choose the AR model when sample sizes are low. Rather the choice between these models should be largely informed by theoretical considerations: The AR model presents a theoretically implausible model in imposing independence between all processes. We have added a discussion of this point to lines 454 - 469:

“Throughout the paper we compared the AR model to the VAR model. However, we believe that it is unnecessarily restricting to choose only between those extremes (all off-diagonal elements zero vs. all off-diagonal elements nonzero). The AR model, by imposing independence between processes, presents a theoretically implausible model for many psychological processes. Applied researchers who estimate the VAR model may be primarily interested in the recovery of cross-lagged effects rather than auto-regressive parameters, for example to determine which processes are dependent on one another (as evidenced by frequent discussions of Granger causality \\cite{granger1969investigating} in these settings). In such settings, one could estimate VAR models with a constraint that limits the number of nonzero parameters or penalizes their size \\cite{fan2001variable, hastie2015statistical}. This would allow the recovery of large off-diagonal elements without the high variance of estimates in the standard VAR model. Similarly, one could estimate a VAR model and, instead of comparing it to an AR model and thus testing the nullity of the off-diagonal elements jointly, test the nullity of the off-diagonal elements of the VAR matrix individually. Further investigation of these alternatives would provide a more complete picture to applied researchers in future studies.”

\\pagebreak

\\textbf{Comment 3}

\\begin{displayquote}

The authors encouraged future studies with more than 6 variables. However, with fewer than 6 variables considered, how would the current findings hold (I reckon n for both estimation and prediction errors likely will go down)? It is likely that it may take fewer n for VAR to outperform AR.

\\end{displayquote}

We agree with the reviewer and would also predict that the errors go down when decreasing the number of variables $p$. While we in the previous version only focused on the case where more variables are included, we now have broadened this discussion to include our predictions for what would happen when fewer variables are included (lines 478 - 485):

“Specifically, we expect that the $n$ at which VAR outperforms AR becomes larger when more variables are included in the model, and smaller when less variables are included. This change may be nonlinear in nature: As we add variables to the model, we would expect the variance of the VAR model to grow much quicker than the variance of the AR model, since in the former case we need to estimate $p^2$ parameters, and in the latter only $p$. However, the bias of the AR model also grows with each new variable added, with $p^2 - p$ elements set to zero in each case, and so again, this will largely depend on the data-generating system at hand. Similarly, we would expect that for models with more variables the 1SER outperforms selecting the model with lowest prediction error for sample sizes larger than 60. While the exact values will change for larger $p$, we expect that the general relationships between $n$, $O$, and $D$ extend to any number of variables $p$.”

\\textbf{Comment 4}

\\begin{displayquote}

For each VAR model (R and D) condition, 100 independent time series were simulated. These are more referred to as “replications” for each model design condition, rather than “iterations” (e.g., page 5 line 148). The authors should revise the term where applicable throughout the manuscript.

\\end{displayquote}

We agree and have changed this term to “replications” throughout. 

\\textbf{Comment 5}

\\begin{displayquote}

The authors simulated n = 500 for estimation simulation but n = 2000 for prediction simulation. From the results and discussions, it appears that 2000 does not matter too much. Discussions are needed regarding this point.

\\end{displayquote}

N = 2000 refers to the size of the test set, which is only used to compute the out-of-sample prediction error. 2000 was chosen to be sufficiently large to yield an accurate estimate of the out-of-sample prediction error (i.e., not subject to sampling variation due to a small test set). This is the quantity which Bulteel et al. approximate using a cross-validation scheme, which adds another source of potential error to their findings - choosing a small test set may yield unreliable estimates of the true out-of-sample prediction error.

For clarity, we have added a simpler description of this to the main text (lines 257 - 265):

“To compute prediction error, we generate a test-set time series consisting of $n_{\\text{test}} = 2000$ observations (using a burn-in of $n_{\\text{burn}} = 100$) for each of the 6000 VAR models described in the previous section. For each of the 100 replications of model and sample size condition, we average over the prediction errors which are obtained when estimated model parameters are evaluated on the test set.”

\\textbf{Comment 6}

\\begin{displayquote}

Figure 4b and on page 11 line 350, the authors should state how many cases have EEcomp unequal to zero.

\\end{displayquote}

It should be noted that the answer to this question depends on how we define “cases”: If we take “cases” to mean all models and sample size conditions (so, 7400 x 493 cases), there is a very low proportion of “cases” in which the two methods pick different models (around 0.5 percent). However, this is not a particularly meaningful metric, since at a large enough sample size, both methods essentially always pick the same (correct) model. We note this latter point explicitly on lines 363 - 365.

“Finally, why does the curve get closer and closer to zero? The reason is that the standard error converges to zero with (the square root of) the number of observations, and therefore the probability that both rules select the same model approaches 1 as $n$ goes to infinity. “

If we instead take “cases” to mean only models, we would ask: For how many of the 7400 models do the 1SE and lowest PE rules choose different models at some n? The answer to this question is that in all but 2 “cases” there is some value of EEcomp unequal to zero. We now note this on line 339-340. 

“Note that for all but 2 of the 7400 models there is some $n$ at which the two decision rules in question choose a different model.”

To reflect this discussion we now more clearly specify the state of affairs regarding cases where the 1SER and lowest prediction error rules differ in the discussion section, and use this to offer additional advice to researchers in practice (lines 413 - 419):

“Our simulations also showed that as $n \\to \\infty$, both decision rules converged to selecting the same model. This means that there is a relatively small range of sample sizes in which these decision rules lead to contradictory model selections for a given data-generating system. We recommend that researchers wishing to use prediction error to choose between these models examine utilize both the 1SER and lowest prediction error rules, and in cases of conflict between the two, use the 1SER for low ($n<60$) sample sizes.”

\\textbf{Comment 7}

\\begin{displayquote}

The authors mentioned mixed models – some recent simulation work on DSEM should be cited, which have shown satisfying estimation results for VAR. Furthermore, the authors should briefly discuss the subgroup/mixture approach when there are distinct subgroups of time series patterns (e.g., GIMME).

\\end{displayquote}

Unfortunately no simulation studies that we are aware of have examined the performance of DSEM in recovering mixed VAR models. The only relevant paper we know of \\cite{schultzberg2018number} is limited to only considering AR(1) models, and the focus of their investigation is on the recovery of fixed effects, rather than individual-specific parameters as is the focus in the current n=1 analyses examined in this paper. 

We agree that GIMME is an interesting approach, but posits a much more general model than considered here (including contemporaneous directed relationships) and again a group-level structure not present in n=1 analyses. We agree however that simulation studies using mixed VAR models and comparing this approach to GIMME would be an interesting future line of research, particularly when the target of inference is the individual-specific parameters. We have extended our discussion of future studies to include this, (lines 513 - 519):

“Finally, it would be useful to study the performance of mixed VAR models in a simulation setting, and perhaps compare this approach to alternative methods of using group-level information in individual time-series analysis, such as GIMME, an approach originally developed for the analysis of brain data [17]. Early simulation studies have assessed the performance of mixed AR models in recovering fixed effects using Bayesian estimation techniques [18], but these analyses have yet to be extended to mixed VAR models or the recovery of individual-specific random effects.”

\\textbf{Comment 8}

\\begin{displayquote}

When referring to the mixed effects examined in Bulteel et al. (2018), at least for the first time (page 2 line 40), it would be helpful to clarify it refers to multilevel model with random effects.

\\end{displayquote}

This is now clarified in text:

“Although the latter statement implies that the estimation error of mixed AR and mixed VAR models are similar, Bulteel et al.[1] conclude that ``[...] it is not meaningful to analyze the presented typical applications with a VAR model'' (p. 14) when discussing both mixed effects (i.e., multilevel models with random effects) and single-subject models.”

\\textbf{Comment 9}

\\begin{displayquote}

On page 4 line 123, it should be Figure 6 in the supplementary materials.

\\end{displayquote}

This has been changed to refer to the Supporting Information throughout

\\textbf{Comment 10}

\\begin{displayquote}

 On page 7 line 201, two “have”s; line 202, two “the”s.

\\end{displayquote}

This has been fixed

\\newpage 

\\Large

\\textbf{Reviewer 2}

\\normalsize

\\textbf{Comment 1}

\\begin{displayquote}

Page 3 and 4: I appreciate the novel methods the authors used to generate their simulated data through the use of parameters, R & D. However, I am concerned that this method introduces artifacts into the sampling scheme, due to the fact that there is a correlation between R & D (as shown in Figure 6). Thus, it seems that there would be bias in the models generated with this technique. In general, although the authors provide some justification for using R & D, it would be helpful for the author to provide further explanation of their parameterization methods in light of this correlation. In particular, it seems that this correlation may be artificially induced by the authors’ definition of R & D. For example, a theorem from linear algebra states that the sum of the eigenvalues of a matrix (i.e., D) is equal to the sum of its diagonal elements (i.e., it’s trace, in this case the AR parameters included in the numerator of R). Hence, the numerator of R is essentially D. This suggests that the R-D parameterization is likely responsible for the correlation in the simulation samples. I recommend that the authors acknowledge this in their description of their parameterization methods. Additionally, I recommend that they examine the correlation between R & D to demonstrate that this correlation is sufficiently low so as to not overly bias the simulation data. Finally, I strongly suggest that authors reformulate R so that it is free from the influences of this correlation, such as by using the current denominator of R. This would allow for the modeling of autoregressive effects (i.e., D) and cross-lagged effects (i.e., denominator of R), independently.

\\end{displayquote}

We thank the reviewer for this comment. On reflection we agree that these were not the optimal dimensions to choose when sampling lagged parameter matrices, for the reasons outlined. We have changed the R dimension to refer to the average absolute cross-lagged parameter value as suggested (now denoted O), and re-ran the simulations accordingly. We have also clarified in text that D should be interpreted as the average auto-regressive parameter (which is equivalent to the average eigenvalue). See lines 102 - 111 (pages 3-4) for changes to the definition, and other changes to the results of our simulation throughout:

“The first characteristic is based on the size of the auto-regressive effects, that is, the absolute values of the diagonal elements of the lagged parameter matrix ($\\Phi_{ii}$) which encode the relationship between a variable and itself at the next time point. We summarize the information contained in these diagonal elements by taking the mean of their absolute values D, given as [...]

Note here that taking the sum of auto-regressive parameters is equivalent to taking the sum of the eigenvalues of $\\Phi$, denoted $\\lambda$. To ensure stationarity, only $\\Phi$ matrices with $|\\lambda| < 1$ are included in our analysis [10]. The second characteristic is based on the size of the cross-lagged parameters ($\\Phi_{ij}, i \\neq j$), encoding the relationships between different processes. We again summarize this information by taking the mean absolute of these parameters, denoted R and given as

[...]

We expect that true VAR models with a high $D$ value and small $O$ value (i.e., large auto-regressive effects and small cross-lagged effects) result in a low estimation error for AR models, since these VAR models are very similar to an AR model. In contrast, if $O$ is high, we expect that the estimation error of the AR model is large, because it sets the large cross-lagged effects in the true VAR model to zero.”

The main results of our paper do not change, though it is now clearer that the mean absolute off-diagonal elements ($O$) largely determines the size of $n_e$. The weighted median $n_e$ is now 89, slightly lower than the value of 92 obtained in the previous simulation. 

We have updated Figure 2 accordingly, and now describe the results as follows (lines 185 - 208):

``Above we suggested that the relative performance of AR and VAR models (quantified by $\\text{EE}_\\text{Diff}$) depends on the characteristics $D$ and $O$ of the true VAR parameter matrix. In Figure 2 (a) we show the median (across models in cells) $n$ at which the estimation error of VAR becomes smaller than the estimation of AR (i.e., $\\text{EE}_\\text{Diff} > 0$). We see that the larger the average off-diagonal elements $O$, the lower the $n$ at which VAR outperforms AR. This is what one would expect: when $O$ is small (as indicated by the lowest rows of cells in Figure 2 (a)), the true VAR model is actually very close to an AR model. In such a situation, the bias introduced by the AR model by setting the off-diagonal elements to zero leads to a relatively small estimation error. This trade-off between a simple model with high bias but low variance and a more complex model with low bias but high variance is well-known in the statistical literature as the \\textit{bias-variance trade-off} \\cite{hastie2009elements}. It therefore takes a considerable amount of observations until the variance of the VAR estimates becomes small enough to outperform the AR model. When $O$ is large (indicated by the upper rows of cells), the bias of the AR model leads to comparatively larger estimation error. Finally, we can also see that the size of the diagonal elements $D$ is not as critical in determining $n_e$ as the size of the off-diagonal elements: Picking any row of cells in Figure 2 (a), we can see that there is only a very small variation across columns, with larger $D$ values appearing to lead to very slight decreases in $n_e$ in general. Note that the $O$ characteristic also largely explains the vertical variation of the estimation error curves shown in Figure 1 (b): the curves on top (small $n_\\text{e}$) have low $O$, while the curves at the bottom (large $n_\\text{e}$) have high $O$. Figure 2 (b) collapses across these values and illustrates the sampling distribution of $n_e$, taking into account the likelihood of any particular VAR matrix (as specified by the mixed model estimated from the ``MindMaastricht'' data).''

\\textbf{Comment 2}

\\begin{displayquote}

 I think it may be useful for the authors to provide more recommendations for the design of psychological time series studies based on their data. In other words, are there suggestions for how applied researchers should implement these findings? For example, do these results support the recommendation of collecting more observations in general? [...] Line 385: In regards to comparing the 1SER rule versus selecting the model with the lower prediction error, what should applied researchers take away from these results if they are working with data with n $>$ 60?

\\end{displayquote}

We should note that our paper largely focuses on the distinction between AR and VAR models in single-subject time series, and the topic of psychological time series studies and methodological design is of course a much broader topic than we can hope to comprehensively address in this paper. However, we can make some rather specific recommendations within the scope of what we have examined. First is that, of course, the average sample size requirement needed for VAR to outperform AR models is n = 89, but this provides only a very rough guidelines for the sample sizes researchers should aim for. Crucially, we see a very large degree of variation around this value, depending on the size of the off-diagonal elements. Thus, knowledge or researcher expectations about the underlying system plays a crucial role in choosing a sufficient sample size. Second, based on our analysis of the 1SER and lowest prediction error decision rules, we can recommend that, in cases where both decision rules pick different models, researchers should use the 1SER for low sample sizes. 

We have made these recommendations more explicit in text, both on lines 411-419:

“In addition, we show via simulation that the 1SER performs better than selecting the model with the lowest prediction error for $n<60$, in cases where those decision rules select conflicting models. Our simulations also showed that as $n \\to \\infty$, both decision rules converge to selecting the same model. This means that there is a relatively small range of sample sizes in which these decision rules lead to contradictory model selections for a given data-generating system. We recommend that researcher wishing to use prediction error to choose between these models examine utilize both the 1SER and lowest prediction error rules, and in cases of conflict between the two, use the 1SER for low ($n<60$) sample sizes.”

And in addition on lines 529- 545:

“To sum up, we used simulations to study the relative performance of AR and VAR models in settings typical for psychological applications. We were able to make clear statements about the average performance of VAR models, which showed that, on average, we need sample sizes approaching $n = 89$ for single-subject VAR models to outperform AR models. While this may seem like a relatively large sample size requirement, such longer time series are becoming more common in psychological research \\cite{wichers2016critical, helmich2020sudden} and mixed models may allow for acceptable performance for shorter time series, though much research on that topic is still required. Importantly, we also found the variance around this average sample size to be considerable, with the variation largely a function of the average absolute value of the off-diagonal (i.e. cross-lagged) effects. Decomposing this variance showed that (i) one cannot expect reliable statements with respect to the relative performance of the AR and VAR models that ignore the characteristics of the generating model, and (ii) that choosing reliably between AR and VAR models is difficult for most sample sizes typically available in psychological research. Finally, we provided a theoretical explanation for when the ``1 Standard Error Rule'' outperforms simply selecting the model with lowest prediction error, and showed that the 1SER performs better when $n$ is small.”

\\textbf{Comment 3}

\\begin{displayquote}

Lines 443-455 refer to several theoretical points about choosing between VAR and AR models under the condition of equal estimation error. Given that applied researchers may want to select one model over the other for hypothesis-testing reasons (e.g., testing the AR effect of mood versus including the cross-lagged effect of anxiety on mood), could you provide clarification on whether an applied researcher would be able to test for estimation error equivalence using empirical data? If that is not possible, I believe it may be helpful to state this explicitly.

\\end{displayquote}

We agree with the point that applied researchers may not necessarily be primarily interested in general estimation error, but instead in, for instance, the ability to correctly identify non-zero cross-lagged effects. This was also a point raised by Reviewer 1. To address this we have added a clarification on this in the discussion (paragraph on choosing between AR and VAR as extremes, lines 447 - 452, see Reviewer 1 comment #1 response for added text) . We suggest alternative approaches if researchers are interested primarily in cross-lagged effects, and possibilities for future studies to investigate this issue.

With regards to testing for estimation error equivalence using empirical data - indeed that is not possible, as it is only possible to try and evaluate prediction error equivalence. We clarify that this is the reason we investigate prediction error in the first place by making changes to lines 225 - 228:

“In the previous section, we directly investigated the estimation errors of the AR and the VAR model in typical psychological applications and showed that the n at which VAR becomes better than AR depends substantially on the characteristics of the true model. In practice, the true model is unknown, so we can neither look up the n at which VAR outperforms AR in the above simulation study, nor can we compute the estimation error on the data at hand. Thus, to select between these models in practice, we may choose to use the prediction error which we can approximate using the data at hand, for instance by using a cross-validation scheme as suggested by Bulteel et al. [1].”

We also address what researchers should do in practice in different sample size conditions at the end of the discussion, which we have outlined in the response to the previous comment of this reviewer.

\\textbf{Comment 4}

\\begin{displayquote}

 Line 173: Could you clarify what is meant by specifying the data generating model and how a researcher would do this using empirical data?

\\end{displayquote}

In this statement we are referring to the results of our simulation study, which show a) that $n_e$ depends substantially on the particular set of lagged parameter values in the data-generating model, b) that the variation in EE across data-generating models is much larger than the variation across replications of the same data-generating model. As such, although it is difficult to make statements about the sample size necessary for the VAR model to outperform the AR model in general, if one has information about the parameters of the data-generating model, one can make much more precise statements about the sample size necessary for the VAR model to outperform the AR model. We have changed this statement to more clearly communicate this (lines 172 - 176):

“However, we see that the sampling variation across replications is smaller than the variation across VAR models for most n. This means that if one has information about the parameters of the data-generating model, one can make much more precise statements about the sample size necessary for the VAR model to outperform the AR model”

Of course, it is not possible to specify the data generating model based on a given empirical dataset: But if researchers are trying to determine an acceptable minimum sample size before data collection, it is probably necessary for them to specify their beliefs about the structure of the data-generating model (such as the expected size of auto-regressive and cross-lagged parameters) to do so in any meaningful way. We explore this further in the analysis which follows the aforementioned statement, for instance in Figure 2 (a).

\\textbf{Comment 5}

\\begin{displayquote}

Line 509: I recommend rephrasing this sentence to specify that the relative performance of AR and VAR models were studied using simulations of data generated from typical psychological applications.

\\end{displayquote}

We agree and this has been changed.

---

## [Decision Letter · Decision Letter 1]

21 Sep 2020

PONE-D-20-03592R1

Choosing between AR(1) and VAR(1) Models in Typical Psychological Applications

PLOS ONE

Dear Dr. Haslbeck,

Thank you for submitting your manuscript to PLOS ONE. After careful consideration, we feel that it has merit but does not fully meet PLOS ONE’s publication criteria as it currently stands. Therefore, we invite you to submit a revised version of the manuscript that addresses the points raised during the review process.

Please, attend the minor suggestion from both reviews.

We look forward to receiving your revised manuscript.

Kind regards,

Miguel Angel Sánchez Granero

Academic Editor

PLOS ONE

Reviewers' comments:

Reviewer's Responses to Questions

**Comments to the Author**

1. If the authors have adequately addressed your comments raised in a previous round of review and you feel that this manuscript is now acceptable for publication, you may indicate that here to bypass the “Comments to the Author” section, enter your conflict of interest statement in the “Confidential to Editor” section, and submit your "Accept" recommendation.

Reviewer #1: All comments have been addressed

Reviewer #2: All comments have been addressed

2. Is the manuscript technically sound, and do the data support the conclusions?

Reviewer #1: Yes

Reviewer #2: Yes

3. Has the statistical analysis been performed appropriately and rigorously? 

Reviewer #1: Yes

Reviewer #2: Yes

4. Have the authors made all data underlying the findings in their manuscript fully available?

Reviewer #1: Yes

Reviewer #2: Yes

5. Is the manuscript presented in an intelligible fashion and written in standard English?

Reviewer #1: Yes

Reviewer #2: Yes

6. Review Comments to the Author

Reviewer #1: The authors are very responsive to my previous comments and have addressed them well. I thank the authors for another contribution to the literature.

One tiny new comment: The authors said on page 8 that "for each of the 6000 VAR models described in the previous section" below "Assessing ngap through simulation." I may have missed it but I only recall the 7400 models the authors mentioned previously.

Reviewer #2: The authors present results from a series of simulation studies examining the performance of AR and VAR models. Results assist the reader in determining which model structure (i.e., AR versus VAR) to use when modeling n=1 time series data. I appreciate the efforts the authors have undertaken to revise the manuscript.

My very minor suggestion is to change “researcher” to “researchers” in line 416.

No further recommendations.

7. PLOS authors have the option to publish the peer review history of their article (what does this mean?). If published, this will include your full peer review and any attached files.

Reviewer #1: No

Reviewer #2: No

---

## [Author Response · Author response to Decision Letter 1]

30 Sep 2020

Dear Editor, 

We are happy to submit a revised version of our manuscript, in which we addressed the two minor comments of the two reviewers. We also made a number of small textual improvements, which did not change any of the content of the manuscript.

Kind regards,

Jonas Haslbeck

---

## [Editor Report · Decision Letter 2]

2 Oct 2020

Choosing between AR(1) and VAR(1) Models in Typical Psychological Applications

PONE-D-20-03592R2

Dear Dr. Haslbeck,

We’re pleased to inform you that your manuscript has been judged scientifically suitable for publication and will be formally accepted for publication once it meets all outstanding technical requirements.

Kind regards,

Miguel Angel Sánchez Granero

Academic Editor

PLOS ONE

Additional Editor Comments (optional):

Please, follow Reviewer 2 suggestion:

My very minor suggestion is to change “researcher” to “researchers” in line 416 (now line 425).

---

## [Editor Report · Acceptance letter]

19 Oct 2020

PONE-D-20-03592R2 

Choosing between AR(1) and VAR(1) Models in Typical Psychological Applications 

Dear Dr. Haslbeck:

I'm pleased to inform you that your manuscript has been deemed suitable for publication in PLOS ONE. Congratulations! Your manuscript is now with our production department. 

Kind regards, 

on behalf of

Dr. Miguel Angel Sánchez Granero 

Academic Editor

PLOS ONE